# ^188^Re-SSS Lipiodol Radioembolization in HCC Patients: Results of a Phase 1 Trial (Lip-Re-01 Study)

**DOI:** 10.3390/cancers15082245

**Published:** 2023-04-11

**Authors:** Etienne Garin, Xavier Palard, Yan Rolland, Samuel Le Sourd, Nicolas Lepareur, Valérie Ardisson, Christelle Bouvry, Sophie Laffont, Boris Campillo-Gimenez, Eric Bellissant, Julien Edeline

**Affiliations:** 1Department of Nuclear Medicine, Cancer Institute Eugène Marquis, F-35042 Rennes, France; 2Campus Santé, University of Rennes, F-35042 Rennes, France; 3INSERM, INRAE, Nutrition Métabolismes et Cancer U1317, University of Rennes, F-35033 Rennes, France; 4CLCC Eugène Marquis, Inserm, LTSI-UMR 1099, University of Rennes, F-35000 Rennes, France; 5Department of Radiology, Cancer Institute Eugène Marquis, F-35042 Rennes, France; 6Department of Medical Oncology, Cancer Institute Eugène Marquis, F-35042 Rennes, France; 7Department of Clinical Research, Cancer Institute Eugène Marquis, F-35042 Rennes, France; 8INSERM CIC 1414 (Clinical Investigation Center), F-35033 Rennes, France; 9Department of Clinical and Biological Pharmacology and Pharmacovigilance, Pharmaco-Epidemiology and Drug Information Center, Rennes University Hospital, F-35033 Rennes, France; 10INSERM, COSS (Chemistry Oncogenesis Stress Signaling)—UMR_S 1242, University of Rennes, F-35042 Rennes, France

**Keywords:** ^90^Y-loaded microspheres, ^188^Re-SSS lipiodol, radioembolization, hepatocellular carcinoma (HCC), radiation therapy

## Abstract

**Simple Summary:**

Radioembolization is a kind of internal radiation therapy which is currently used in non-operable HCC. Several radiolabeled product can be used as ^90^Y-loaded microspheres or less frequently radiolabeled lipiodol either with iodine 131 (in the past) or currently with rhenium 188 (^188^Re). Currently used ^188^Re-labeled compounds are limited by in vivo instability. This study sought to evaluate the safety, bio-distribution, and response to ^188^Re-SSS lipiodol, a new and more stable compound. Method: Lip-Re-01 was an activity-escalation Phase 1 study involving HCC patients progressing after sorafenib. The primary endpoint was safety. Secondary endpoints included bio-distribution and response evaluation. Results: Fourteen heavily pre-treated HCC patients were treated in Level 1 (*n* = 6), Level 2 (*n* = 6), and Level 3 (*n* = 2). Safety was acceptable with only 1/6 of Level 1 and 1/6 of Level 2 patients experiencing limiting toxicity. The study was prematurely discontinued unrelated to clinical outcomes. Cumulative urinary elimination and fecal eliminations at 72 h were very low, confirming high in vivo stability. Partial response occurred in 37.5% of the patients receiving 3.6 GBq of ^188^Re-SSS lipiodol or more. Conclusion: As the 3.6 GBq activity proved to be safe, it will be used in an ongoing Phase 2 study.

**Abstract:**

Background: Despite the wide development of ^90^Y-loaded microspheres, ^188^Re-labeled lipiodol is still being used for radioembolization of hepatocellular carcinoma (HCC). However, the use of this latter compound is limited by in vivo instability. This study sought to evaluate the safety, bio-distribution, and response to ^188^Re-SSS lipiodol, a new and more stable compound. Method: Lip-Re-01 was an activity-escalation Phase 1 study involving HCC patients progressing after sorafenib. The primary endpoint was safety based on Common Terminology Criteria for Adverse Events (AEs) of Grade ≥3 within 2 months. Secondary endpoints included bio-distribution assessed by scintigraphy quantification from 1 to 72 h, tumor to non-tumor uptake ratio (T/NT), as well as blood, urine and feces collection over 72 h, dosimetry, and response evaluation (mRECIST). Results: Overall, 14 heavily pre-treated HCC patients were treated using a whole liver approach. The mean injected activity was 1.5 ± 0.4 GBq for activity Level 1 (*n* = 6), 3.6 ± 0.3 GBq for Level 2 (*n* = 6), and 5.0 ± 0.4 GBq for Level 3 (*n* = 2). Safety was acceptable with only 1/6 of Level 1 and 1/6 of Level 2 patients experiencing limiting toxicity (one liver failure; one lung disease). The study was prematurely discontinued unrelated to clinical outcomes. Uptake occurred in the tumor, liver, and lungs, and only sometimes in the bladder. The T/NT ratio was high with a mean of 24.9 ± 23.4. Cumulative urinary elimination and fecal eliminations at 72 h were very low, 4.8 ± 3.2% and 0.7 ± 0.8%, respectively. Partial response occurred in 21% of patients (0% in the first activity level; 37.5% in the others). Conclusion: The high in vivo stability of ^188^Re-SSS lipiodol was confirmed, resulting in encouraging responses for a Phase 1 study. As the 3.6 GBq activity proved to be safe, it will be used in a future Phase 2 study.

## 1. Introduction

Hepatocellular carcinoma (HCC) is the most common primary liver cancer, being the third leading cause of cancer-related death worldwide, with around 745,000 deaths reported annually [1]. Despite the use of ^90^Y-loaded microsphere radioembolization and immunotherapy, such as the atezolizumab + bevacizumab regimen [2], the treatment of patients with locally advanced HCC with a Barcelona Clinique Liver Cancer (BCLC) classification of B, as well as of those with portal vein thrombosis (PVT), is still challenging. Indeed, given this patient population, the overall response rate is only 35% with immunotherapy [2] and 50% with ^90^Y-loaded microsphere radioembolization based on personalized dosimetry [3], whereas median overall survival (OS) values in the same studies were 19.2 [2] and 26.6 months, respectively [3]. In addition, randomized studies comparing ^90^Y-loaded resin microspheres versus sorafenib did not demonstrate any increased OS upon using radioembolization [4,5,6]. In this context, the development of new therapeutic agents appears mandatory to further improve HCC prognosis. Concerning radioembolization, the use of new radiolabeled compounds of lipiodol is still of interest.

Historically, ^131^I lipiodol has been in use for many years, and the agent was approved for PVT patients after an improved overall survival (OS) benefit demonstrated in a randomized study involving HCC patients with PVT as compared with best supportive care [7]. Nevertheless, the agent’s marketing was discontinued in 2010 on account of several reasons, such as product drawbacks (mainly strong radioprotection constraints due to using iodine 131 and lung toxicities, in some cases), along with the emergence of new alternatives, including the approval of sorafenib and development of ^90^Y-loaded microspheres. At the same time, a large field of research focused on developing lipiodol radiolabeling with isotopes other than iodine 131, such as rhenium 188 (^188^Re).

^188^Re has proven to be an isotope suitable for internal radiation therapy with a β^−^ emission of high energy (2.12 MeV), on account of its short half-life of 17.9 h, with only a small amount of gamma emission suitable for imaging, yet with a lower energy compared to ^131^I lipiodol (155 keV and abundance of 14% versus 364 keV and abundance of 81.7% for iodine 131), thereby resulting in lower radioprotection constraints.

The current labeling is based on the synthesis of a lipophilic complex of ^188^Re, which is secondarily solubilized into lipiodol. Several lipiodol labeling attempts with ^188^Re have been reported [8,9,10], the only one still in clinical use being the ^188^Re-4-hexadecyl-1-2,9,9-tetramethyl-4,7-diaza-1,10-decanethiol lipiodol (^188^Re-HDD lipiodol) [10]. Nevertheless, the main drawback of such labeling turns out to be a low in vivo stability, with about 44% of the injected activity being eliminated in urine 46 h post-injection [11].

Given this context, we have developed a new radiolabeling of lipiodol with ^188^Re, namely the ^188^Re-(S_2_CPh)(S_3_CPh)_2_ lipiodol (^188^Re-SSS lipiodol) [12,13], which is characterized by a very high in vivo stability in animals with less than 5% of urinary excretion at 48 h post-injection [12]. Automatization of its synthesis has been developed, as well [14].

The primary objective of the study was to evaluate the safety of ^188^Re-SSS lipiodol in an activity-escalation study involving non-operable HCC patients progressive after sorafenib. Secondary objectives included bio-distribution analysis, dosimetry analysis, and preliminary response evaluation. The full results of the Lip-Re-1 study are reported here, though the preliminary bio-distribution results concerning the first six treated patients have been previously reported [15].

## 2. Methods

### 2.1. Patients

Eligible patients were compliant with the following criteria: HCC diagnosed based on either histological criteria or the European Association of Study of the Liver (EASL) non-invasive criteria [16], which were non-resectable, non-transplantable, non-accessible to percutaneous ablation, and progressive after sorafenib; the tumor was to be measurable, being either uni- or multinodular, involving less than 50% of hepatic volume, being Stage A to C of the BCLC classification, with the patient age of 18 years or older; World Health Organization (WHO) performance status score 0–2, the patient being in progression after sorafenib (or having contraindication to receiving sorafenib or intolerance to sorafenib). The exclusion criteria were as follows: Stage ≥3 toxicity of the CTCAE version 4.03, Stage D of the BCLC classification, acute impairment of hepatic functions (Child–Pugh B9 or C), Grade III HCC of the Okuda classification, encephalopathy, advanced chronic respiratory insufficiency, creatinine clearance <55 mL/min, polynuclear neutrophils <1500 G/L, platelets <50 G/L, prothrombin <40% (INR > 2), contraindication to intra-arterial administration, urinary incontinence, and other progressive cancer. Patients who could not be followed (for psychological or geographic reasons), patients dependent on another person for daily care, pregnant or breastfeeding women, or patients not using an adequate effective contraception method were also excluded. Based on trial amendments, 3 other exclusion criteria were added: Child–Pugh classification ≥ B8, DLCO < 60%, lung shunt based on ^99m^Tc macro-aggregated albumin (MAA) planar scintigraphy > 20%.

All patients provided written informed consent before enrolment. Ethical approval for this study was obtained from CPP Ouest V ethical committee (number 09/18-730-18.07.27.53027). The study was undertaken in accordance with the Declaration of Helsinki principles and respective amendments. The study was registered at ClinicalTrials.gov under number NCT01126463 and the EudraCT register under number 2009-013231-37.

### 2.2. Study Design

This monocentric Phase 1 clinical trial was designed as a standard (3 + 3) open-label activity-escalation study, starting with an activity of 1.85 GBq of ^188^Re-SSS lipiodol and level increment of 1.85 GBq until a planned activity of 7.4 GBq. Each level included 3 patients. If no limiting toxicity occurred, the next level was authorized. If one limiting toxicity occurred, three additional patients were included in the same level and, if no supplemental limiting toxicity occurred, the next level was authorized, otherwise dose escalation was interrupted. The study stopped for toxicity reasons if more than one limiting toxicity occurred in any activity level of three or six patients.

An Independent Safety Committee Review Board (ISCRB) was met regularly to monitor study conduct and safety concerns if required.

All patients received intra-arterial injection of ^188^Re-SSS lipiodol into the hepatic artery under local anesthesia using the classical Seldinger technique, with a whole liver approach, as classically applied with ^131^I lipiodol. ^188^Re-SSS lipiodol was locally prepared as previously described [10,12,13].

The patients were hospitalized in a dedicated radionuclide therapy room for three days following ^188^Re-SSS lipiodol administration, in order to conduct the bio-distribution analysis. Further follow-up comprised physical examinations, clinical chemistry assessments (including electrolytes, renal and liver function tests, aFP), and hematological tests at 24 h, 48 h, and 72 h post-administration, every month for 4 months, with a triphasic contrast-enhanced abdominal CT performed at 4, 8, and 16 weeks after treatment.

### 2.3. Assessment of Adverse Events and Activity-Limiting Toxicities

Any clinical or laboratory adverse event was recorded and scored according to the National Cancer Institute’s CTCAE, version 4.03. Accountability related to the studied treatment was assessed following ICH E2B (R3) guidance, meaning that for patients with both liver toxicity and evidence of largely progressive disease, toxicity was attributed to disease progression (and not to the treatment). An activity-limiting toxicity was defined as a permanent Grade 3 or higher toxicity observed within two months following the ^188^Re-SSS lipiodol administration, resulting in contraindication to a second injection.

The maximum tolerated activity (MTA) was defined as the maximal injected activity resulting in no more than zero or one limiting toxicity in three or six patients of the same activity level, respectively.

### 2.4. Bio-Distribution Assessment

#### 2.4.1. Image Acquisition

Whole body planar acquisitions (256 × 1054 matrix), thoraco-abdominal planar acquisitions (256 × 256 matrix), and thoraco-abdominal single-photon emission computed tomography combined with computed tomography (SPECT/CT) acquisitions (three ordered-subset expectation maximization, 32 projections, 180°, 128 × 128 matrix, five iterations, eight subsets with a Gauss filter, 4.8 mm/pixel) were performed at 1, 6, 24, 48, 72 h post-administration, using a Symbia T2 gantry (Siemens Healthcare,) equipped with high-energy parallel-hole collimators. The acquisition window was 155 keV (20%). The SPECT images were reconstructed using attenuation (low-dose CT based), dead time, and scatter corrections.

#### 2.4.2. Quantitative Analysis

The geometric mean of anterior and posterior measurements was evaluated for planar acquisitions. Upon SPECT/CT study, volumes of interests (VOIs) were drawn around the liver (including tumor), tumor, lungs, and background region to calculate the total amount of activity in these volumes. Volume and activity in the healthy liver were calculated by subtraction of the liver and tumor parameters. T/NT uptake ratio was also calculated in abdominal SPECT studies (using a 3 cm^3^ VOI positioned on the higher-uptake area of the tumor and on the surrounding healthy liver).

#### 2.4.3. Blood, Urine, and Feces Collection

Total urine and feces emissions were collected during the hospitalization. Blood was sampled at 1 h, 6 h, 12 h, 24 h, 48 h, and 72 h after treatment for ^188^Re content measurements. The total activity (in % of administrated activity, % AI) was extrapolated by considering a blood volume of six liters and hematocrit of 40%. Samples were analyzed in a gamma counter calibrated for ^188^Re (Packard Bioscience Cobra II model 5002).

#### 2.4.4. Dosimetry

Dosimetry was evaluated according to the medical internal radiation dose (MIRD) formalism, taking into account the biological elimination in urine and feces, as previously described.

#### 2.4.5. Tumor Response Assessment

Tumor response assessment on contrast-enhanced abdominal CT was evaluated at 2 and 4 months according to the Response Evaluation Criteria in Solid Tumors version 1.1. (RECIST 1.1), while using the modified RECIST method. Serum alpha fetoprotein (AFP) measurement at baseline, M2, and M4 was also performed.

#### 2.4.6. Statistical Analysis

Any patient who received one injection of ^188^Re-SSS lipiodol was included in the analysis population whichever the purpose was, i.e., the safety, bio-distribution, dosimetry, or efficacy assessment. Quantitative values were expressed as means +/− standard deviation, while counting values were described by absolute and relative frequencies. One additional analysis—not initially planned—was also conducted to describe overall survival of the cohort. Survival times were measured from the date of inclusion to the date of death from any cause or the date of last known news at the investigator site. The survival curve was estimated by the Kaplan–Meier method and presented with a 95% two-sided confidence interval using the log–log transformation. Statistical analyses were performed using SAS software^®^, version 9.4.

## 3. Results

Overall, 14 heavily pre-treated HCC patients with progressive disease after sorafenib were treated between June 2010 and March 2019, with six pertaining to activity Level 1 (1.85 GBq), six to activity Level 2 (3.7 GBq), and two to activity Level 3 (5.55 GBq). Mean tumor size was 8.9 ± 5.5 cm, with 85.7% of the patients displaying multifocal disease and 35.7% portal vein thrombosis. Global patient characteristics are presented in Table 1. The mean injected activity was 1.5 ± 0.4 GBq for Level 1, 3.6 ± 0.3 GBq for Level 2, and 5.0 ± 0.4 GBq for Level 3. Finally, recruitment was prematurely stopped unrelated to clinical outcomes but due to the very low accrual rate observed in the course of the study due to the change in the treatment landscape of HCC, and the MTA was not reached.

### 3.1. Safety

Two limiting toxicities were reported, including one of six patients in the first activity level and one of six patients in the second activity level. No limiting toxicity was observed among the two patients of the third activity level.

The reported limiting toxicities consisted of one liver failure in Level 1 and one lung injury in Level 2.

The patient with liver failure presented with a huge, 10 cm, rapidly progressing tumor, with liver function deterioration starting during the screening period prior to treatment and very rapidly progressive disease after treatment. Liver failure was considered to relate to disease progression by both the investigators and ISCRB, whereas the ISCRB decided conservatively to classify this event as limiting toxicity, given that the death occurred before the time point evaluation of 2 months. On account of this event, a substantial modification of the protocol was performed. It concerned adding liver function of Child–Pugh B8 into the exclusion criteria (only Child–Pugh ≥ B9 was excluded in the first study protocol) and liver function checking so as to validate eligibility criteria within 48 h before ^188^Re-SSS lipiodol injection.

The patient with lung failure had pre-existing lung disease whose severity had probably been misdiagnosed. Therefore, this limiting toxicity was rather deemed to be related to inaccurate patient selection. Two substantial protocol modifications were implemented. The first one consisted of introducing functional respiratory testing with exclusion of patients exhibiting a DLCO value < 60%. The second one introduced a more accurate tool to evaluate the risk of lung injuries related to SIRT, currently performed with ^90^Y-loaded microsphere SIRT: lung shunting evaluation based on macro-aggregated albumin (MAA) quantification with exclusion of patients displaying a lung shunt > 20%. Following these two amendments, no other lung disease of any grade was reported in the study.

Considering all adverse events (AEs), whatever the grade or treatment accountability, there were 166 AEs reported (Table 2). They were of Grades 1–5 in 46%, 33%, 17%, 1%, and 3% of cases, respectively. Five AEs of Grade 5 were recorded, including the two limiting toxicities and three death cases related to cancer progression which occurred after the time point of 2 months. At least one AE of Grades 1–5 was observed in 100%, 86%, 64%, 14%, and 36% of patients, respectively.

Treatment-related AEs of Grade ≥ 3 were reported in eight (57%) patients, including the liver failure declared as limiting toxicity, the lung injury also declared as limiting toxicity, alanine aminotransferase increase, aspartate aminotransferase (AST) increase, and lymphopenia (Table 3). The most common treatment-related AEs of Grade ≥ 3 occurring in a least 10% of patients were AST increase observed in 21% of patients and lymphopenia observed in 28.6% of patients.

### 3.2. Bio-Distribution

Bio-distribution analysis with whole body acquisition found uptake only in the tumor, liver, lungs, and sometimes in the bladder, with the exception of one patient where gastric uptake was also detected. Based on SPECT/CT evaluation, mean whole liver, tumor, and lung uptakes were, respectively, 84.4 ± 10.1%, 51.5 ± 21.3%, and 14.7 ± 9.7% of the detected activity at 1 h; 86.0 ± 10.1%, 56.0 ± 20.2%, and 13.8 ± 10.1% at 6 h; 81.9 ± 11.2%, 51.0 ± 20.5%, and 17.0 ± 11.3% at 24 h; 82.3 ± 11.9%, 52.4 ± 22.5%, and 16.2 ± 11.7% at 48 h; 79.3 ± 12.8%, 50.9 ± 21.7%, and 20.5 ± 12.7% at 72 h (Figure 1). Blood activity was 0.91 ± 0.3% of injected activity at 1 h, 1.68 ± 1.1% at 6 h, 2.0 ± 0.9% at 24 h, 2.24 ± 1.15% at 48 h, and 2.84 ± 1.8% at 72 h. Cumulative urinary elimination and fecal eliminations at 72 h were only 4.8 ± 3.2% and 0.7 ± 0.8% of the injected activity, respectively.

### 3.3. Dosimetry

Mean absorbed doses were, respectively, 9.7 ± 2.9 Gy for the whole liver, 37.9 ± 35.0 Gy for tumor, 3.7 ± 2.1 Gy for healthy liver, 2.0 ± 1.5 Gy for lungs for the 1.85 GBq activity level; 18.6 ± 6.4 Gy for the whole liver, 62.8 ± 27.7 Gy for tumor, 23.4 ± 21.9 Gy for healthy liver, 4.8 ± 2.2 Gy for lungs for the 3.7 GBq activity level; 29.6 ± 12.8 Gy for the whole liver; 162.9 ± 46.5 Gy for tumor, 18.6 ± 8.5 Gy for healthy liver for lungs for the 5.55 GBq activity level. Dosimetry by GBq injected was 5.8 ± 2.1 Gy/GBq to the whole liver, 23.9 ± 19.3 Gy/GBq to the tumor, 4.6 ± 5.1 Gy/GBq to the healthy liver, and 1.3 ± 0.8 Gy/GBq to the lungs.

### 3.4. Tumor Response

Based on RECIST criteria, partial response was observed in two of fourteen patients (14%), with no response in the first activity level (zero of six), and two responses in the eight patients (25%) of the subsequent activity level (one response in each level), as shown in Figure 2. Based on mRECIST criteria, partial response was observed in three of fourteen patients (21%) with no response in the first activity level (zero of six), and three responses in the eight patients (37.5%) of the subsequent activity level (two of six in Level 2 and one of two in Level 3). The three imaged responders also showed a normalization of their AFP level from 392.9 at baseline to 7 UI/mL, 20.7 to 7, and 17.4 to 3.7.

### 3.5. Subsequent Treatment of ^188^Re-SSS Lipiodol

Two patients received, outside the study, a compassionate second treatment of ^188^Re-SSS lipiodol, with neither of them experiencing any limiting toxicity 2 months following the second treatment. The first patient was included in activity Level 1 and received a second injection of 3.1 GBq at 4 months (cumulative activity of 4.6 GBq), due to liver progression at 4 months following a tumor stabilization at 2 months. Two months after the second injection, the patient was still progressing and died 10.5 months after the first treatment. The second patient was included in activity Level 2 and received a second injection of 2.2 GBq at 4 months (cumulative activity of 5.6 GB), due to a good but incomplete partial response so as to maximize the response. Two months after the second treatment, the patient was in complete mRECIST response. He finally died of disease progression 29.6 months after the first treatment.

### 3.6. Overall Survival

After a maximum follow-up of 59 months, 13 (93%) out of the 14 treated patients died, resulting in a median overall survival estimated at 8.5 months (95% CI: [4.6–14.5]). Overall survival curve and 6th, 12th, and 24th month rates are presented in Figure 3.

## 4. Discussion

Safety of ^188^Re-SSS lipiodol used in a whole liver treatment was proven acceptable in this Phase 1 study up to the planned activity of 5.55 GBq. It must be underlined that the two limiting toxicities recorded were most likely due to inaccurate patient selection owing to insufficiently restrictive inclusion criteria at the time of trial design. The patient with liver failure, which was considered conservatively to be a limiting toxicity despite clear tumor progression, was, in fact, included with a Child–Pugh B9 status; such inclusion was, at that time, still permitted by the inclusion criteria. Currently, the limit of Child–Pugh B7 is well accepted. The second patient with lung injuries displayed a history of pre-existing lung disease, which was probably insufficiently documented. In the aftermath of this observation, MAA lung shunt evaluation was proposed to be added to the inclusion criteria, based on the use of ^90^Y- or ^166^Ho-labeled microspheres. The most commonly reported Grade ≥3 toxicities were aspartate aminotransferase increases (21%) and lymphopenia (28.6%), as commonly observed with labeled microspheres [3,17].

Furthermore, two patients received activity Level 3 with 5 GBq injected, while two other patients received a secondary injection with cumulative injected activities of ^188^Re-SSS lipiodol of 4.6 and 5.6 GBq, without limiting toxicity.

Considering bio-distribution, this study clearly confirms the better stability, at high activity levels, of ^188^Re-SSS lipiodol in comparison with ^188^Re-HDD lipiodol, with a very low urinary excretion of less than 5% versus 44% for ^188^Re-HDD lipiodol [11]. This high stability of the ^188^Re-SSS complex may be due to an oxidation level of +III of ^188^Re within this complex. While this is more difficult to achieve, it is also chemically more stable with a lower oxidation risk in Levels +V and +VII. In ^188^Re-HDD, complex ^188^Re is in oxidation Level +V with a high risk of oxidation in Level +VII, with release of ^188^Re in serum. This point is a great advantage for ^188^Re-SSS lipiodol. Based on this high stability, along with the high tumor targeting level observed in this study, ^188^Re-SSS lipiodol appears to be the most favorable radiolabeling of lipiodol with ^188^Re available today.

Dosimetry values turned out to be difficult to analyze in a Phase 1 activity-escalation study. Therefore, comparison among studies is similarly difficult, on account of the highly different activities being used, either in the same study or among studies. Standardization of absorbed doses by the activity injected enables a more accurate dosimetry comparison. For tumors, the mean tumor absorbed dose was 23.9 Gy/GBq injected with ^188^Re-SSS lipiodol, in comparison with the 15.9 Gy/GBq injected reported in one study with ^188^Re-HDD [18]. In our study, the normal liver mean dose was 4.6 Gy/GBq injected, and the lung mean dose was 1.3 Gy/GBq injected. For activity Level 2, the mean tumor dose was about 62 Gy. This value is lower than the recognized tumoricidal dose of 100–120 Gy for ^90^Y resin [19] spheres and 205 Gy for ^90^Y glass spheres [19]. Nevertheless, radiobiology of these latter products is different, and radiobiology of ^188^Re-SSS lipiodol is not well documented. One study that assessed the impact of tumor dose on overall survival reported survival to be significantly improved for patients receiving a tumor dose higher than 30 Gy [18].

Considering efficacy, the response rate for the global population was only 14% with RECIST and 21% with mRECIST. Median OS was 8.5 months (95% CI: [4.6–14.5]), slightly lower than the median OS described with a systemic drug used in second line treatment after sorafenib of 10.6 months (95% CI: [9.1–12.1]) with regorafenib [20] and 10.2 months (95% CI: [9.1–12.0]) with cabozantinib [21] in slightly different populations (PVT in studies with systemic drugs).

However, it has to be underlined that the lower activity level (1.85 GBq) used in this Phase 1 activity escalation was probably under-dosed as no response was observed (i.e., associated with a tumor dose lower than the tumoricidal dose of ^188^Re-SSS lipiodol). This point means that 43% of the patients were under-dosed, with a negative impact on efficacy. Indeed, in patients treated with higher activities, the response rate was 25% with RECIST and 37.5% with mRECIST. In comparison, yet in slightly different populations, the overall response rate was 35% with atezolizumab + bevacizumab [2] and 50% with ^90^Y glass microspheres using personalized dosimetry [3].

The whole liver approach did more than likely impact the response rate observed, and it is well recognized that the more selective the fixed activity, the higher the absorbed dose, and the better the response rate observed. This is well documented with the concept of radiation segmentectomy described first for ^90^Y glass microspheres, where the mean segmental dose was 521 Gy, while injecting, at a segmental level, the activity required to deliver a mean absorbed dose of 120 Gy for an injection at the lobe level [22]. Using a lobar or even segmental approach for further studies, ^188^Re-SSS lipiodol will most likely optimize efficacy, as described with ^90^Y-loaded microspheres, while using a segmental approach versus a lobar approach. Indeed, the reported response rates were 80 to 90% for segmental treatments [22,23] versus only about 50% for lobar treatments [3,17]. Targeting smaller lesions than the ones that have been targeted in this Phase 1 study appears to be another option to increase efficacy. Using the criteria of the European Association of Study of the Liver, Shinto et al. reported a 90% response rate, in a study with a mean lesion size of 6.6 cmin, in comparison with 8.9 cm in Lip-Re-1, while using a mean activity of 2 GBq of ^188^Re-HDD lipiodol [10].

The use of ^188^Re-radiolabeled lipiodol offers several potential advantages in comparison with radiolabeled microspheres. First, the mechanism of uptake differs with microembolization in microarteries as observed with loaded microspheres, but also with the possibility of radiolabeled lipiodol to deeply enter into the vasculature, at the sinusoid capillary level, resulting in a more homogeneous distribution [12,24], while being incorporated into tumor cells themselves [25]. On account of these bio-distribution differences, both the efficacy and toxicity profile between radiolabeled lipiodol and radiolabeled microspheres may actually differ. As an example, no risk of either gastric ulcer or cholecystitis has been reported to date when using radiolabeled lipiodol despite whole liver treatment without any coiling procedure. Second, ^188^Re is produced by a generator, ^188^W/^188^Re, with a prolonged half-life of 69 h, thus allowing for delivery of ^188^Re over several months in radiopharmacy and at a relatively low cost of about EUR 50,000. The latter could be of particular interest regarding the global cost of radioembolization. This is especially true in developing countries, where using ^90^Y-loaded microspheres cannot be afforded. Third, ^188^Re-radiolabeled lipiodol may exert an adjuvant action, as demonstrated with ^131^I lipiodol in two randomized trials [26,27] and in a meta-analysis [28], where the hazard ratio of OS was 0.5 for patients receiving ^131^I lipiodol [28]. Indeed, the adjuvant setting may actually become a new area of development for ^188^Re-radiolabeled lipiodol.

One of the main limitations of this well-conducted Phase 1 activity-escalation study was the low patient recruitment. This led to considerable extension of the study duration and to a premature end to the study, unrelated to clinical outcomes, while the MTA was not identified. Finally, there were multiple reasons for making the decision to stop the study early but they were never related to safety concerns; extra cost to maintain the generator, COVID-19 shutdown, and modification to the standard of care were the main ones. Nevertheless, we argue that the recommended activity for further trials could be limited to 3.7 GBq. This level of activity is already quite high and within a range enabling comparison with the mean activity level usually used with ^90^Y-loaded microspheres, meaning 1.4 GBq with resin microspheres [4] and 3.6 GBq with glass microspheres using personalized dosimetry [3]. The second main limitation was that the study focused on activity escalation, instead of liver absorbed dose escalation, with this concern also being evident with ^90^Y-loaded microspheres. For this reason, the maximal normally tolerated liver absorbed dose is still unknown, and it must thus be evaluated in further studies. In this regard, a future Phase 2 trial evaluating efficacy of ^188^Re-SSS lipiodol with planned activity level at 3.7 GBq in intermediate and advanced HCC patients, sponsored by the Cancer Institute Eugène Marquis and funded by the annual French Institutional Grant against Cancer (n° PHRC-K19-161), is already in the process of starting with the first inclusions expected during 2023.

## 5. Conclusions

The safety of ^188^Re-SSS lipiodol used in a whole liver treatment was proven acceptable in this Phase 1 study. ^188^Re-SSS lipiodol was associated with a favorable bio-distribution for radioembolization, especially exerting the highest in vivo stability of any radiolabeled lipiodol compound described to date. The MTA was not identified but the activity of 3.7 GBq could be considered as the recommended nominal activity for further studies. A Phase 2 clinical trial using the activity of 3.7 GBq is currently scheduled since national funding was obtained.

## Figures and Tables

**Figure 1 cancers-15-02245-f001:**
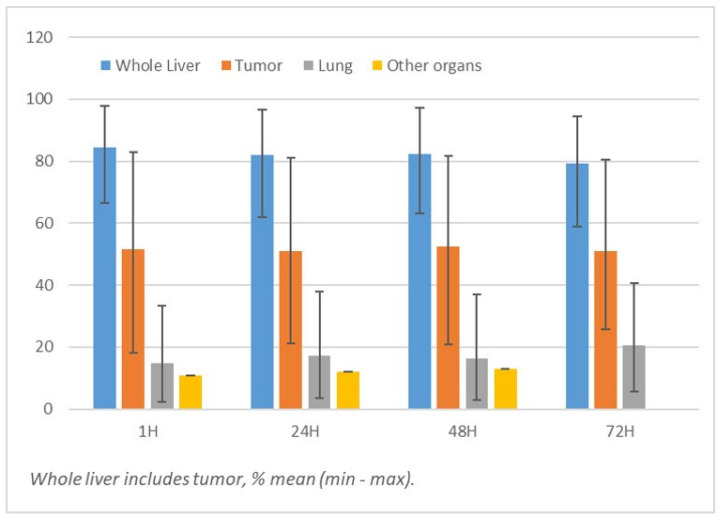
^188^Re-SSS lipiodol SPECT/CT.

**Figure 2 cancers-15-02245-f002:**
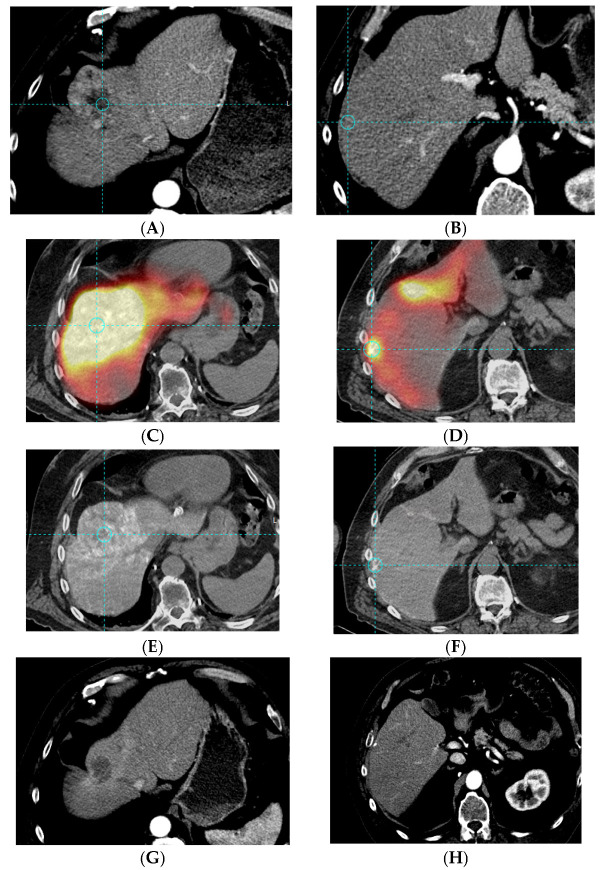
Case of a 70-year-old patient, BCLC B, with progressive disease after sorafenib, Child–Pugh A5, ECOG 0, treated with 3.44 GBq of ^188^Re-SSS lipiodol using a whole liver approach. Baseline CT scan: lesion of 7 cm of segment VIII and IV ((**A**), dotted line and circle), second lesion of 1 cm of segment V/VI ((**B**), dotted line and circle), no PVT ^188^Re-SSS lipiodol SPECT/CT at 24 h: very high tumor uptake (**C**,**D**) and high lipiodol retention (**E**,**F**) of the main lesion and of the small satellite lesion, as well four months post-injection: good partial response of the main lesion (**G**) and complete response (**H**) of the satellite lesion (with only residual lipiodol retention). Global overall survival was 31 months after SIRT.

**Figure 3 cancers-15-02245-f003:**
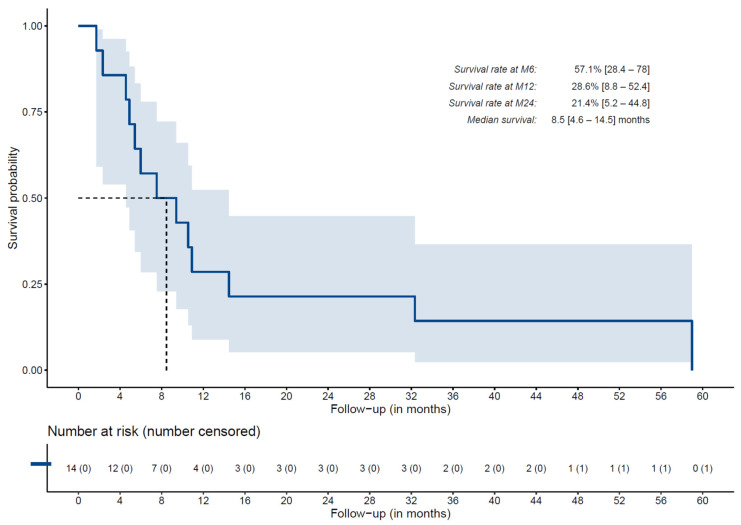
Overall survival after at least one injection of ^188^Re-SSS lipiodol in the 14 heavily pre-treated HCC patients with progressive disease after sorafenib.

**Table 1 cancers-15-02245-t001:** Baseline patient characteristics.

	*n* = 14
Age, mean ± SD (years)	71 ± 4
Gender	
Female	1 (7.1%)
Male	13 (92.9%)
Child–Pugh liver function classification	
A5	9 (64.3%)
A6	3 (21.4%)
B7	1 (7.1%)
B8	1 (7.1%)
ECOG performance status	
0	9 (64.3%)
1	5 (35.7%)
BCLC classification	
B	9 (64.3%)
C	5 (35.7%)
Portal vein invasion	
Present	5 (35.7%)
Absent	9 (64.3%)
Cirrhosis	
Present	13 (92.9%)
Absent	1 (7.1%)
Treatment line	
Second	6 (42.9%)
Third	5 (35.7%)
Fourth	3 (21.4%)
Tumor distribution	
Unifocal	2 (14.3%)
Multifocal	12 (85.7%)
Lobes affected	
Unilobar disease	4 (28.6%)
Bilobar disease	10 (71.4%)
Index tumor size, mean ± SD (cm)	8.9 ± 5.5
AFP level, mean ± SD (kU/L)	3371.5 ± 12282
Bilirubin level, mean ± SD (μmol/mL)	19.6 ± 11.6

**Table 2 cancers-15-02245-t002:** Summary of all adverse events in the analysis population.

	Events				Patients			
Characteristic	Overall	1.85 GBq Level	3.7 GBq Level	5.5 GBq Level	Overall	1.85 GBq Level	3.7 GBq Level	5.5 GBq Level 3
Grade								
1	76 (46%)	25	43	8	14 (100%)	6	6	2
2	54 (33%)	17	33	4	12 (86%)	5	6	1
3	29 (17%)	12	11	6	9 (64%)	4	3	2
4	2 (1%)	1	1	0	2 (14%)	1	1	0
5	5 (3%)	2	2	1	5 (36%)	2	2	1

**Table 3 cancers-15-02245-t003:** Related adverse events of Grade ≥ 3 according to the activity level.

Characteristic	Overall, *n* = 10 ^1^	1.85 GBq Level, *n* = 2 ^1^	3.7 GBq Level, *n* = 5 ^1^	5.5 GBq Level, *n* = 3 ^1^
Preferred Term				
Acute respiratory failure	1 (10%)	0 (0%)	1 (20%)	0 (0%)
Alanine aminotransferase increased	1 (10%)	0 (0%)	1 (20%)	0 (0%)
Aspartate aminotransferase increased	3 (30%)	1 (50%)	2 (40%)	0 (0%)
Hepatic failure	1 (10%)	1 (50%)	0 (0%)	0 (0%)
Lymphopenia	4 (40%)	0 (0%)	1 (20%)	3 (100%)

^1^* n* = number of events, % = % of the events in relation to the total number of events.

## Data Availability

Protocol, SAP, data, and all safety reports are available at Cancer Institute Eugène Marquis, Rue Flandre Dunkerque, CS44229, 35042, Rennes Cedex, France.

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
