# Peer review of "188Re-SSS Lipiodol Radioembolization in HCC Patients: Results of a Phase 1 Trial (Lip-Re-01 Study)"

_cancers, 2023, doi:10.3390/cancers15082245_

Round 1

Reviewer 1 Report

My raitnig of the presented article is not the best, due to the fact that as  a surgeon treating HCC, I would expect more excellent qualification for treatment, safer profile of the method used and slightly better results. But I have no formal objections to the presented study

Author Response

Dear reviewer, thank you for your evaluation and your comments.

I perfectly agree with you, the eraly stop was very unfortunate but we really tried to go as far as possible.

As no specific queries have been raised by all reviewers, I have not modified this article (except based on the editor query asking for a more clear statement that the early stoppage was unrelated to clinical outcome).

Best regards

Reviewer 2 Report

New and useful paper.

Questions: 

1. What is the main question addressed by the research? 2. Do you consider the topic original or relevant in the field? Does it address a specific gap in the field? 3. What does it add to the subject area compared with other published material? 4. What specific improvements should the authors consider regarding the methodology? What further controls should be considered? 5. Are the conclusions consistent with the evidence and arguments presented and do they address the main question posed? 6. Are the references appropriate? 7. Please include any additional comments on the tables and figures.

Answers:
  1. The main question addressed by the research is the method of radioembolization of hepatocellular carcinoma.
  2. The topic is certainly original and address a specific gap in this field.
  3. New addition included clinical evaluation of relatively new, rarely used method of radioembolization.
  4. The authors showed clinical results of Lipiodol-Radioembolization as alternative of Yttrium-Radioembolization. It can be useful in cases when Yttrium is not available or contraindicated.
  5. The conclusions are consistent with the evidence and presented arguments; they address the main question posed.
  6. References are appropriate.
  7. The tables and figures are appropriate and properly show the data. They are easy to understand and interpret. The statistics is good.  

Author Response

(The authors gave the same response as above.)

Reviewer 3 Report

There is a very nice comparison in radioactive amount in a modern technique to treat a rare tumor (HepatoCell Carcinoma). The research is in phase 1 study and can be a good guide for next steps of investigations in this field.

1. What is the main question addressed by the research? 2. Do you consider the topic original or relevant in the field? Does it address a specific gap in the field? 3. What does it add to the subject area compared with other published material? 4. What specific improvements should the authors consider regarding the methodology? What further controls should be considered? 5. Are the conclusions consistent with the evidence and arguments presented and do they address the main question posed? 6. Are the references appropriate? 7. Please include any additional comments on the tables and figures.

1- What is the best amount of radioactive for Rhenium to use in treatment of HCC?

2- There are not so many articles in this field. so I am sure it can complete some gaps.

3- More exact dose delivery in a physicist point of view.

4- As we are faced with a very rare tumor in this manuscript, the dividing the patients to 3 groups can be a good idea.

5- For a phase 1 research it is reasonable, in next steps they should be more careful.

6- It seems yes.

I have no more time to talk about this manuscript, so if you need more, you can omit my name from the reviewers list.

Author Response

(The authors gave the same response as above.)

Reviewer 4 Report

This manuscript demonstrated that 188Re-SSS lipiodol radioembolization with 3.6 radioactivity was safe in a phase I dose escalation study. The study protocol set 3 levels; Level 1 (1.85GBq), Level 2(3.7 GBq) and Level 3 (5.55 GBq). This phase I study was well organized and authors well summarized the results. However, it was unfortunate that the study was discontinued during the Level 3 stage although the MTA was not reached. Authors describe the following reasons why the study was stopped; cost, COVID-19, and change of treatment landscape of HCC. However, these reasons were unclear and also there was a possibility of leading different conclusions if the study finished completely. Again, it was very unfortunate that this study was prematurely stopped.

Author Response

(The authors gave the same response as above.)

Round 2

Reviewer 4 Report

This reviewer better understands this issue.